# Enhancing Collaborative Medical Outcomes through Private Synthetic Hypercube Augmentation: PriSHA

**Shinpei Nakamura-Sakai[1], Dennis Shung[2†], Jasjeet Sekhon[13†]**

[1] Department of Statistics and Data Science
[2] Yale School of Medicine
[3] Department of Political Science
Yale University, New Haven, Connecticut, USA
{s.nakamura.sakai, dennis.shung, jasjeet.sekhon}@yale.edu

## Abstract

Effective collaboration across medical institutions presents a significant challenge, primarily due to the imperative of maintaining patient privacy. Optimal machine learning models in healthcare demand access to extensive, high-quality data to achieve generality and robustness. Yet, typically, medical institutions are restricted to data within their networks, limiting the scope and diversity of information. This limitation is especially pronounced in the case of patients with rare or unique characteristics, resulting in decreased accuracy for this minority group. To address these challenges, our work introduces a framework designed to enhance existing clinical foundation models, Private Synthetic Hypercube Augmentation (PriSHA). We leverage generative models to produce synthetic data, generated from diverse sources, as a means to augment these models while adhering to strict privacy standards. This approach promises to broaden the dataset's scope and improve model performance without compromising patient confidentiality. To our knowledge, our framework is the first synthetic data augmentation framework that merges privacy-preserving tabular data and real data from multiple sources.

## Introduction

A primary challenge for machine learning models for clinical decision support is ensuring data privacy. Clinical models take Electronic Health Record (EHR) data as input variables, which may contain both direct and indirect identifiers that can be used to link the patient to their protected health information (PHI). The Health Insurance Portability and Accountability (HIPAA) law (Fitzgerald 2015) specifically defines 18 identifiers that must be removed, and the nationally accepted standard is no greater than 0.04% reidentification risk (Emam 2013). Synthetic data is a promising solution to maintain patient privacy while enabling wider use of previously sensitive data. Synthetic data has been used for a wide array of applications across finance, satellite images, and healthcare. (Jordon et al. 2022) (Giuffrè and Shung 2023) Synthetic data effectively combats data scarcity by creating datasets imbued with characteristics that are useful for downstream analysis; these characteristics are fidelity, which is the degree of resemblance in distribution between the synthetic and original datasets and utility, representing the applicability and effectiveness of the data in specific tasks. Most importantly, synthetic data maintains privacy, protecting sensitive information in the original dataset from being exposed.

While synthetic data presents a valuable solution for research and analysis when access to real data is restricted by privacy concerns, several drawbacks exist. As highlighted by (Hittmeir, Ekelhart, and Mayer 2019), synthetic data can result in reduced downstream performance (Manousakas and Aydore 2023) further notes a lack of substantial evidence supporting the usefulness of synthetic tabular data for augmentation. Additionally, in a systematic review by (Hernandez et al. 2022), various studies focus on data augmentation or privacy preservation in medical tabular data, yet none tackle the issue synthetic data augmentation using privacy-preserving synthetic data from multiple data sources.

Moreover, different hospitals may experience distribution shifts, which could be mitigated through data sharing. However, the imperative need to protect patient privacy makes this infeasible. Therefore, it is crucial to develop and implement frameworks aimed at maximizing performance augmentation using synthetic data while maintaining privacy guarantees, ensuring the optimal utilization of synthetic data in healthcare contexts.

### Our contributions

1. Our approach effectively adapts synthetic data to datasets with varying distributions, taking advantage of distribution shifts.

2. Our approach, employing supervised learning, uses select synthetic data subsets to specifically target and enhance downstream task performance. This method outperforms traditional data augmentation, which often fails to capture the heterogeneity of data, by adeptly managing and leveraging this diversity for more effective results.

3. Our methodology not only surpasses standard augmentation performance but also ensures privacy, thereby showing a statistically significant improvement. This approach facilitates collaborative medical outcomes by enabling the sharing of private synthetic data to bolster machine-learning models.

---

[†]Denotes co-senior authorship.

# Private Synthetic Data Generation

For differentially private synthetic data generation we deploy two Generative Adversatial Network (GAN) based methods, Differentially Private GAN (DPGAN) (Xie et al. 2018) and Private Aggregation of Teacher Ensembles-GAN (PATE-GAN) (Jordon, Yoon, and van der Schaar 2018). DP-GAN and PATE-GAN are both machine learning frameworks designed to generate synthetic data while ensuring the privacy of individuals in the training dataset. DPGAN integrates differential privacy into the traditional GAN structure by adding noise to the gradients, ensuring that the final model does not reveal sensitive information about the training data. This approach, however, requires a careful balance between data utility and privacy. On the other hand, PATE-GAN employs a different strategy based on the PATE framework (Papernot et al. 2018). It uses an ensemble of teacher models, each trained on a disjoint subset of the original data, to generate labels for a student model. The student model, which is a GAN in the case of PATE-GAN, then learns to generate synthetic data based on these labels. The PATE framework ensures that the student model's learning process is differentially private, as it only has access to aggregated information from the teacher models, significantly reducing the risk of exposing sensitive information from the training dataset.

# Synthetic Data Augmentation from Multiple Data Sources

In healthcare analytics, recognizing and adjusting for distribution shifts in EHR is paramount, as these shifts profoundly influence the precision and dependability of machine learning models used in clinical decision-making. Such shifts can be attributed to a variety of factors, including evolving patient demographics, changes in clinical practices, or alterations in how data is documented. These changes can substantially compromise the efficacy of models initially trained on EHR data, making it crucial to continually adapt these models. Ensuring models remain generalizable across diverse healthcare contexts, upholding high standards of data integrity, and adhering to the stringent ethical and regulatory demands of the healthcare sector are essential.

To conceptualize a specific scenario, envision two datasets: one from Hospital A, denoted as $\mathcal{D}_A = \{\mathcal{Y}_A, \mathcal{X}_A\}$, and another from Hospital B, $\mathcal{D}_B = \{\mathcal{Y}_B, \mathcal{X}_B\}$, each with distinct distributions where $\mathcal{Y}_* \in \mathbb{R}^{M_*}$ and $\mathcal{X}_* \in \mathbb{R}^{M_* \times p}$ and $M_*$ and $p$ indicates number of observation and covariates. Suppose our objective is to predict outcomes for $\mathcal{D}_A^{pred}$, a dataset of patients more closely aligned in characteristics with $\mathcal{D}_B$ than with $\mathcal{D}_A$. Specifically, $\mathcal{D}_A^{pred}$ refers to a dataset within Hospital A where the model trained using only $\mathcal{D}_A$ struggles to make accurate predictions. This challenge arises because the data in $\mathcal{D}_A^{pred}$ represents a minority distribution within Hospital A, indicating that the model's ability to predict outcomes for this subset is not as strong as for the majority distribution. However, the predictive performance on $\mathcal{D}_A^{pred}$ could potentially be improved by incorporating data from $\mathcal{D}_B$, which may provide additional insights or represent similar minority distributions from another context, thereby enriching the model's training data and enhancing its accuracy for the challenging subset within Hospital A.

In an initial approach, we might consider utilizing solely $\mathcal{D}_A$ for our model, $\hat{\mu}_{\mathcal{D}_A} = f(\mathcal{Y}_A \sim \mathcal{X}_A)$, where $f(\mathcal{Y} \sim \mathcal{X})$ symbolizes a regression motivated estimator (though $f$ could represent any machine learning estimator), and $\hat{\mu}$ is the resultant learned function.

However, in an ideal scenario devoid of privacy constraints, we could incorporate $\mathcal{D}_B$, forming a combined dataset

$$\mathcal{D}_{AB} = \begin{pmatrix} \mathcal{D}_A \\ \mathcal{D}_B \end{pmatrix} = (\mathcal{Y}_{AB}, \mathcal{X}_{AB})$$

and then fit $\hat{\mu}_{\mathcal{D}_{AB}} = f(\mathcal{Y}_{AB} \sim \mathcal{X}_{AB})$. This model is expected to perform better, benefiting from the generalizability gleaned from $\mathcal{D}_A$.

The focus of this manuscript, however, is on an alternative scenario where, due to data privacy concerns, we cannot directly use $\mathcal{D}_B$. Instead, we generate a differentially private synthetic dataset $S_B(N; \epsilon, \delta)$ from $\mathcal{D}_B$, where $S_B$ includes $N$ observations, formulated under privacy constraints $\epsilon$ and $\delta$ defined in definition 1. Typically, this $N$ is the number of observation of $\mathcal{D}_A^{pred}$. The dataset for model training then becomes

$$\mathcal{D}_{AB'} = \begin{pmatrix} \mathcal{D}_A \\ \mathcal{S}_B(N; \epsilon, \delta) \end{pmatrix} = (\mathcal{Y}_{AB'}, \mathcal{X}_{AB'}),$$

and we fit $\mu(\mathcal{D}_{AB'}) = f(\mathcal{Y}_{AB'} \sim \mathcal{X}_{AB'})$. This approach leverages data from Hospital B with a defined level of privacy assurance. The underlying hypothesis is that this model, while potentially less informative than $\mu(\mathcal{D}_{AB})$ using real data, should still surpass the performance of $\mu(\mathcal{D}_A)$, as it incorporates information from the patient group of interest, albeit with data that has been modified for privacy considerations.

# Private Synthetic Hypercube Augmentation (PriSHA)

Building on the insights of (Manousakas and Aydore 2023), which underscore the limited evidence for the utility of synthetic tabular data in model training enhancement, our PriSHA framework adopts a supervised approach. It augments models using only the most predictive hypercube of synthetic data, based on the premise that the most beneficial attributes of synthetic data are concentrated within a specific portion of the dataset. To enable this selective integration, PriSHA employs Bayesian Optimization.

The PriSHA framework begins by computing the feature importance for each covariate in the dataset, employing any established metric from the literature to identify the most significant contributors to the model's predictive power. Based on this analysis, it selects a specific number of covariates or dimensions of the hypercube, denoted by $K$ which can be any natural number. The final step involves optimizing the intervals for each chosen dimension using Bayesian optimization to determine the most relevant synthetic data for augmentation, thereby enhancing the dataset

with precision-targeted information for improved model performance. Further details on each of these steps will be discussed in the remainder of this section, providing a deeper understanding of the methodology and its application. To clarify this point, Figure 2 can be compared to Figure 1.

The initial phase of our methodology involves estimating the feature importance for each covariate within the dataset. In machine learning, the significance of features can be evaluated using a variety of techniques, we can utilize any feature importance metric identified in the literature, a choice that will be further discussed in the subsequent section. The selection of the $K$ parameter is crucial in our analysis, significantly influencing the segmentation and control over the synthetic data. A smaller $K$ yields a more constrained data segmentation, thereby reducing the level of granularity control. In contrast, a larger $K$ complicates conditional sampling due to the increased computational demand for acquiring specific data samples. The probability of sampling a constrained observation for the $i^{th}$ variable is denoted as $\mathbb{P}(\mathcal{X}_i \in [\alpha_i, \alpha_i + \beta_i])$. For $K$ covariates, this probability is expressed as $\prod_{i=1}^{K} \mathbb{P}(\mathcal{X}_i \in [\alpha_i, \alpha_i + \beta_i])$, and this value decreases quickly as $K$ increases.

Following feature importance, we proceed to selectively sample the synthetic data. This process is guided by the hyperparameters $\boldsymbol{\alpha} \in \mathbb{R}^k$ and $\boldsymbol{\beta} \in \mathbb{R}^k_+$. Here, $\boldsymbol{\alpha}$ represents the starting point of the interval from which we sample. In contrast, $\boldsymbol{\beta}$ determines the length of this interval, focusing on the top $k$ features as indicated by their feature importance. This approach results in the generation of a 'sliced' $k$-dimensional hypercube of the synthetic data tailored by $[\alpha_i, \alpha_i + \beta_i] \quad \forall i \in \{1, ..., k\}$. The selection of $\boldsymbol{\alpha}$ and $\boldsymbol{\beta}$ is achieved by solving the following Bayesian optimization problem with uniform prior:

$$
\begin{aligned}
\boldsymbol{\alpha^*}, \boldsymbol{\beta^*} = \underset{\boldsymbol{\alpha} \in \mathbb{R}^k, \boldsymbol{\beta} \in \mathbb{R}^k_+}{\arg\min} \quad & \mathcal{L}(\mathcal{Y}_A^{val}, \hat{\mathcal{Y}}_A^{val}) \\
\text{s.t} \quad & D_{B'} = S_m(N; \epsilon, \delta) \\
& \mathcal{D}_{AB'} = \begin{pmatrix} \mathcal{D}_A \\ \mathcal{D}_{B'} \end{pmatrix} \quad (1) \\
& \hat{\mu}_{AB'} = f(\mathcal{Y}_{AB'} \sim \mathcal{X}_{AB'}) \\
& \hat{\mathcal{Y}}_A^{val} := \hat{\mu}_{AB'}(\mathcal{X}_A^{val})
\end{aligned}
$$

where $\mathcal{L}$ is the desired loss function of interest and $\mathcal{Y}_{val}$ is the outcome on the validation set

These hyperparameters define a targeted hypercube or specific slice of data. By focusing on this segment, we aim to bolster downstream performance through a strategic combination of real data and privately generated synthetic data. This approach is designed to optimize the augmentation process, ensuring that the integration of synthetic data is not only seamless but also maximally beneficial to the overall model's effectiveness. The specifics of the algorithm are outlined in Algorithm 1, which delineates the process for generating the synthetic data utilized in data augmentation.

---

**Algorithm 1: PriSHA**

1: **Input:** $N, k, \mathcal{D}_A, \mathcal{D}_A^{pred}, \mathcal{D}_B, \epsilon, \delta$
2: Split $\mathcal{D}_A^{pred} = \{\mathcal{D}_A^{val}, \mathcal{D}_A^{test}\}$
3: Calculate feature importance scores from $f(\mathcal{Y}_A \sim \mathcal{X}_A)$
4: Initialize $\boldsymbol{\alpha}^0 = \{\alpha_i\}_{i \in \{1,...,k\}}$ and
5: $\boldsymbol{\beta}^0 = \{\beta_i\}_{i \in \{1,...,k\}}$
6: Train $S_B(N; \epsilon, \delta)$ using $\mathcal{D}_B$
7: **for** $j = 1, \ldots, J$ **do**
8: $\quad$ Conditionally sample $D_{B'}^j = S_m(N; \epsilon, \delta)$ with constraint

$$\mathcal{X}_i \in [\alpha_i, \alpha_i + \beta_i] \quad \forall i \in \{1, ..., k\}$$

9: $\quad$ Create $\mathcal{D}_{AB'}^j = \begin{pmatrix} \mathcal{D}_A \\ \mathcal{D}_{B'}^j \end{pmatrix}$
10: $\quad$ Train $\hat{\mu}^j = f(\mathcal{Y}_{AB'}^j \sim \mathcal{X}_{AB'}^j)$
11: $\quad$ Estimate $\hat{Y}_{val}^j = \hat{\mu}^j(\mathcal{X}^{val})$
12: $\quad$ Compute $l^k = \mathcal{L}(\hat{Y}_{val}^k, \mathcal{Y}_{val})$
13: $\quad$ Suggest $\boldsymbol{\alpha}^{j+1}$ and $\boldsymbol{\beta}^{j+1}$ using Bayesian optimization based on $\{\boldsymbol{\alpha}^0, ..., \boldsymbol{\alpha}^j\}, \{\boldsymbol{\beta}^0, ..., \boldsymbol{\beta}^j\}$, and $\{l^0, ..., l^j\}$
14: **end for**
15: **return** $D_{B'}^{j^*}$ where $j^* = \arg\min_k l^j$

---

## Experimental Results

To evaluate the performance of PriSHA, we conducted a series of analyses using EHR data of patients who presented to a tertiary academic health center with acute gastrointestinal bleeding from 2014 to 2019, which is the most common gastrointestinal condition requiring hospitalization. Input variables include demographics, measuring vital signs, initial laboratory values, and nursing assessments. The outcome is a binary composite outcome of whether or not the patient received a hospital-based intervention (red blood cell transfusion, intervention to stop bleeding, or all-cause 30-day mortality). We used metadata of race/ethnicity to create two subsets: patients who identified themselves as ethnically Hispanic and those who did not. These subsets were denoted as $\mathcal{D}_A$ and $\mathcal{D}_B$, respectively. Although various predictive models were available, we opted for XGBoost (Chen and Guestrin 2016) due to its state-of-the-art performance on tabular data (Borisov et al. 2022) (Shwartz-Ziv and Armon 2021). The effectiveness of the model was gauged using the Area Under the Receiver Operating Characteristic curve (AUC) as the loss function. For additional details about the dataset, please see Table 2.

Our initial investigation aimed to determine the potential benefits of augmenting our dataset. To this end, we established two baseline AUC metrics: the first AUC was calculated for predictions on $\mathcal{D}_A^{Pred}$ using a model exclusively trained on $\mathcal{D}_A$, while the second AUC was derived from $\mathcal{D}_A^{Pred}$ using a model trained on the combined dataset $\mathcal{D}_{AB'}$. A comparison of these AUC values, facilitated by the DeLong test (DeLong, DeLong, and Clarke-Pearson 1988), revealed that the augmented model significantly outperformed its non-augmented counterpart, as detailed in Table 3. Therefore, we established a performance threshold: any augmen-

| Method | $\varepsilon$ | Standard AUC (%) | PriSHA AUC (%) | t-stat | p-val |
|---|---|---|---|---|---|
| DPGAN | 0.1 | 85.45 | 85.52 | 0.3782 | 0.7095 |
| DPGAN | 1.0 | 85.31 | 85.34 | 0.3156 | 0.7557 |
| DPGAN | 2.0 | 85.14 | 85.70 | 2.8583 | 0.0101** |
| DPGAN | 3.0 | 85.25 | 85.50 | 1.5105 | 0.1474 |
| DPGAN | 5.0 | 85.04 | 85.35 | 1.9779 | 0.0626* |
| DPGAN | 10.0 | 85.26 | 85.50 | 1.1372 | 0.2696 |
| DPGAN | 20.0 | 85.45 | 85.70 | 1.6071 | 0.1245 |
| DPGAN | 100.0 | 85.44 | 85.47 | 0.1623 | 0.8728 |
| PATE-GAN | 0.1 | 85.60 | 85.33 | -1.4450 | 0.1647 |
| PATE-GAN | 1.0 | 84.94 | 85.28 | 1.2849 | 0.2143 |
| PATE-GAN | 2.0 | 83.90 | 85.68 | 4.2384 | 0.0004*** |
| PATE-GAN | 3.0 | 82.79 | 85.64 | 5.6358 | ≤0.0001*** |
| PATE-GAN | 5.0 | 85.53 | 86.27 | 2.5170 | 0.0210** |
| PATE-GAN | 10.0 | 85.62 | 86.39 | 1.97 | 0.0638* |
| PATE-GAN | 20.0 | 85.51 | 86.39 | 3.7618 | 0.0013*** |
| PATE-GAN | 100.0 | 85.51 | 86.22 | 2.8335 | 0.0106** |

Table 1: Standard Augmentation versus PriSHA using PATE-GAN and DPGAN across various $\epsilon$ values. Significance levels are indicated as $^*p < 0.10$, $^{**}p < 0.05$, $^{***}p < 0.01$

tation resulting in an AUC below $85.3\%$ would be considered detrimental. At the same time, an AUC exceeding $88.87\%$ represents a benchmark that is challenging to achieve with the addition of private synthetic data alone.

In this study, feature importance was assessed using gain and weight. Although gain is commonly preferred, our dataset reveals that the top two features are lab_HGT and lab_HCT, corresponding to hemoglobin and hematocrit levels, respectively. These two features are highly correlated, with a 95% correlation coefficient, reflecting their shared focus on red blood cell metrics. This high correlation suggests that selecting both features might lead to redundancy in feature representation, as gain tends to create repetitive divisions based on these two similar features. Furthermore, given the nature of boosting algorithms, features utilized after hemoglobin and hematocrit primarily serve to rectify errors from preceding trees, which predominantly involve these two covariates. Contrastingly, lab_PLT shows considerably lower correlations with hemoglobin and hematocrit (3% and 6%, respectively), thereby capturing an orthogonal component that the former two do not predict. Consequently, we opted to use weight over gain for determining feature importance, as it better represents diverse and distinct aspects of our dataset. See figure 3 for more detail.

In our study, we determined that the variable 'gain' is predominantly influenced by hemoglobin levels, as demonstrated in Figure 3. Therefore, we selected $k = 3$, leading to the formation of a standard cube. This decision strikes a balance between adequate control over data segmentation and the maintenance of a manageable level of complexity in conditional sampling. For the Bayesian optimization process to suggest optimal values for $\alpha$ and $\beta$, we employed the hyperopt package (Bergstra, Yamins, and Cox 2013).

We proceeded to evaluate the performance of standard synthetic data augmentation, achieved by simply integrating $\mathcal{D}_A$ with $S(N; \epsilon, \delta)$, against that of PriSHA. To ensure robustness, 20 experiments were conducted, and a t-test was performed for each method and $\epsilon$ value. We focused on two methods, DPGAN and PATE-GAN where we used the synthcity and we fixed the $\delta$ to be $\frac{1}{M\sqrt{M}}$ as suggested by (Qian, Cebere, and van der Schaar 2023) and explored a range of $\epsilon$ values: $0.1, 1, 2, 3, 5, 10, 20, 100$. The results are presented in Table 1.

Our analysis revealed that DPGAN generally enhanced performance but not significantly. In contrast, PATE-GAN showed a significant improvement in downstream performance for $\epsilon$ values greater than 1. For extremely low $\epsilon$ values, such as 0.1, the synthetic data might become excessively noisy, lacking sufficient informative content for effective model integration. The optimal AUC was obtained using PATE-GAN with $\epsilon = 10, 20$, although these values still do not reach the ideal AUC of 88.87% achievable with non-private data augmentation. However, they do surpass the 85.30% AUC noted in scenarios without any augmentation.

## Discussion and Future Work

This paper has presented an approach to synthetic data augmentation in healthcare, particularly focusing on enhancing collaboration from multiple hospitals while maintaining privacy.

The key strength of our approach lies in its ability to selectively utilize the most predictive hypercube of synthetic data. Our experimental results confirm that this targeted approach to data augmentation can significantly improve model performance, especially in contexts where privacy concerns restrict the use of real-world data.

In future research, we aim to refine our model by incorporating the meta-learning approach as discussed in (Hamad et al. 2023). This method would allow us to combine the strengths of DPGAN and PATE-GAN to produce enhanced private synthetic data.

**Institutional Review Board (IRB)** The study is approved by the IRB with protocol number: 1408014519

## Definition

We use the widely used differential privacy to ensure the privacy of the generated synthetic data introduced in (Dwork, Roth et al. 2014).

**Definition 1 (Differential Privacy)** *A randomized algorithm $\mathcal{M}$ is $(\epsilon, \delta)$-differential private if for all $\mathcal{S} \subset \mathcal{O}$ and for all $\mathcal{D}$ and $\mathcal{D}'$ which differs only on a single observation:*

$$\mathbb{P}(\mathcal{M}(\mathcal{D})) \leq e^{\epsilon} \, \mathbb{P}(\mathcal{M}(\mathcal{D}') \in \mathcal{S}) + \delta$$

where $\mathcal{O}$ is the output space,

## Data Description

Table 2: Observation counts, number of features, and outcome distribution across demographic groups in EHR dataset.

| Dataset | Dem Grp | Outcome | Obs | Feat | Label=1 |
|---------|---------|---------|-----|------|---------|
| EHR | Hispanic | Composite | 599 | 38 | 31.55% |
| | Non-Hisp. | | 3723 | | 39.70% |

## Big picture of PriSHA

In Figures 1 and 2, we present an overview of standard data augmentation techniques alongside PriSHA.

## Evaluating the Presence of Distribution Shifts

For "Hispanic Only", the AUC is 85.30%, and for "Hispanic + Non-Hispanic", it is higher at 88.87%.

DeLong's test is used to compare the AUCs statistically. The p-value of 0.0167 suggests that the difference in AUC between the two data types is statistically significant. This implies that the model's performance in distinguishing between classes is significantly better when both Hispanic and Non-Hispanic data are included compared to when only Hispanic data is used.

Table 3: Comparison of AUC Values

| Data Type | AUC (%) | DeLong's p-value |
|-----------|---------|------------------|
| Hispanic Only | 85.30 | 0.0167 |
| Hispanic + Non-Hispanic | 88.87 | |

## Feature Importance

The blue bars represent the Normalized Gain, which reflects how much each variable contributes to the model's predictive power by measuring the average improvement in accuracy or reduction in loss when a variable is used in the model.

The green bars represent the Normalized Weight, which indicates how frequently each variable is used in splitting the data across all trees within a model like Random Forest or Gradient Boosting.

Variables are sorted by their Normalized Weight in descending order, meaning those at the top of the plot are used more frequently in model decisions, while those at the bottom are used less frequently.

## Experimental Results: Standard Augmentation versus PriSHA

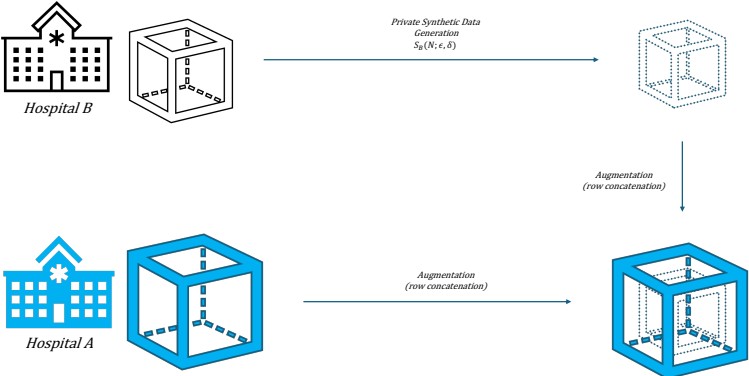

Figure 1: Overview of Standard Augmentation

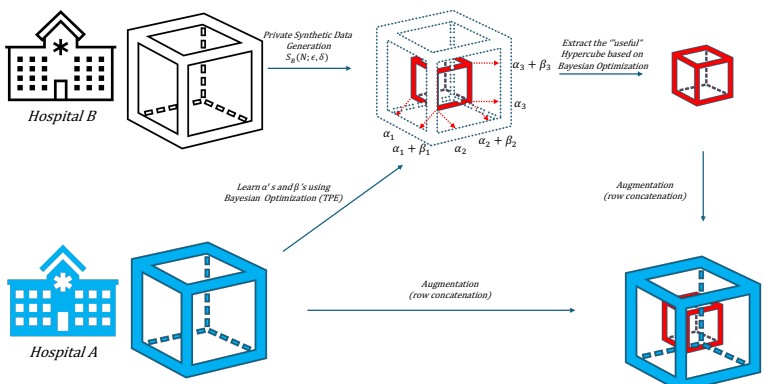

Figure 2: Overview of PriSHA

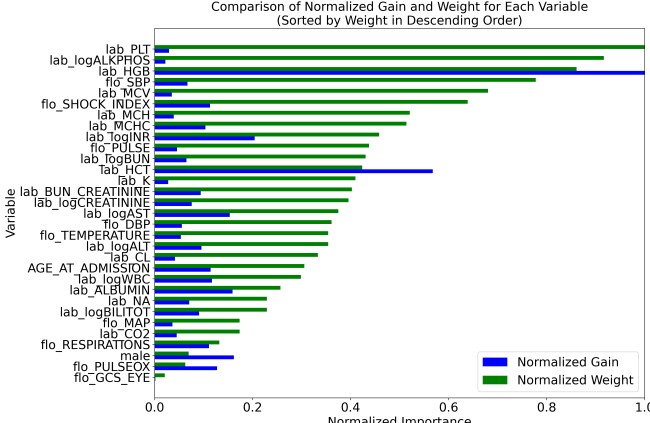

Figure 3: Feature importance of electronic health record data used for the predictive model. These include demographic variables (age, biological sex), initial laboratory values (metabolic chemistries, blood counts, and liver function tests), nursing assessments (Glasgow Coma Score [GCS]), and vital sign measurements (blood pressure, respiratory rate, temperature, pulse). The top three features are laboratory values: platelet count, alkaline phosphatase, and hemoglobin level.

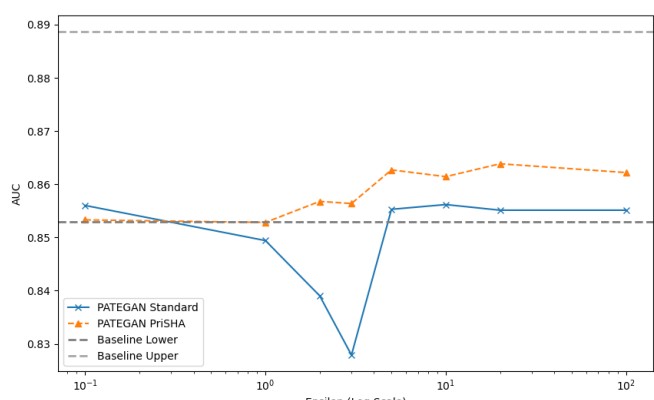

Figure 5: Average External Validation AUC Over 20 Simulations: PATE-GAN in PriSHA vs. Standard Augmentation

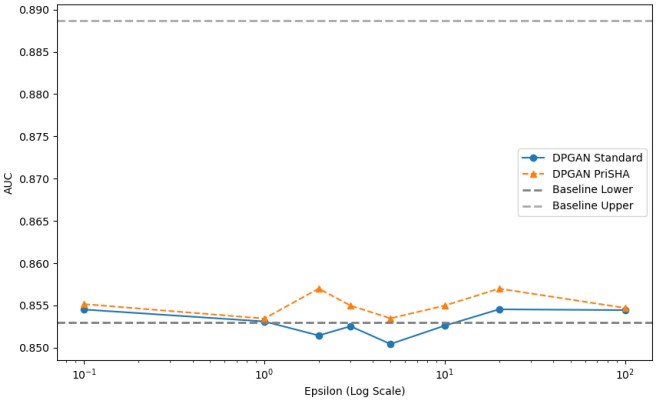

Figure 4: Average External Validation AUC Over 20 Simulations: DPGAN in PriSHA vs. Standard Augmentation

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
