# OpenReview forum: "Enhancing Collaborative Medical Outcomes through Private Synthetic Hypercube Augmentation: PriSHA"
_AAAI.org/2024/Spring_Symposium_Series/Clinical_FMs — AAAI 2024 SSS on Clinical FMs_

### Official Review · Reviewer_a2Rx · 2024-02-14

**Rating:** 7
**Confidence:** 3

**Review:**

The paper proposes a privacy-preserving synthetic data generation technique PriSHA for assisting training clinical foundation models. The DPGAN and PATE-GAN are combined to make the synthetic data generation private, enhancing the privacy protection capability. The experiments show slight improvement of PriSHA over PATEGAN Standard.

Strength:
1. Privacy protection and using synthetic data is very important in healthcare.
2. The combination of existing techniques makes sense.
3. The experimental results are surprisingly good enough.

Opportunity to improve:
1. I am not sure if I buy in the concept that the proposed method can "address distribution shift". To me, it is just combining two data sources, except that one is synthetic. It is quite different from deliberate design of tackling distribution shift. It would connect more dots to federated learning.
2. I understand DPGAN provides noises to protect training data, while PATE-GAN trains student models to generate synthetic data for model training. I am puzzled why the improvement is not over standard DPGAN, but over the PATE-GAN. More importantly, I do not think synthetic data can significantly improve model performance in general, since it is impossible to generate OOD data out of the scope of the existing data distribution to provide new information. This should be not be the main focus of the study.
3. There is no evaluation and real-world cases showing whether the privacy is protected well enough. This is the main motivation of this paper, but it is poorly presented and rarely discussed.

---

### Official Review · Reviewer_Ar9n · 2024-02-20
**Applying generative models to encourage data sharing is very interesting**

**Rating:** 8
**Confidence:** 3

**Review:**

# Review to 17
Strength:
- This paper studies the problem of generating medical data while preserving privacy. This is an interesting and important problem.
- The idea of combining Bayesian optimization and privacy-preserving generative models to generate informative and differentially private data is novel.

Weakness:
- The optimization specification and pseudo-code is a bit hard to follow. Here are some questions/problems:
-- definition of $K$
-- I assume $\mathcal{X}_i$ is the $i$-th dimension of the input space. Is my understanding correct? If yes, please define the notation.
-- When sampling from $[\alpha_i, \alpha_i+\beta_i]$, I assume a uniform distribution is used. Is this correct? If yes, please specify.
- The empirical results are not so impressive.
-- It seems that the data augmentation, both standard and PriSHA, can only improve the AUC marginal (85.30%);
-- When DPGAN is selected, the benefit of including Bayesian optimization, which inevitably increases computation complexity, is not so significant (only 2 out of 8).

Problems
- The benchmark with 88.87% AUC is claimed to be obtained using $D_{AB'}$. Is this distribution $D_{AB}$?
- Could you provide some details of the used dataset? For instance, how many samples are included? What is the ratio of different ethnical groups?

---

### Official Review · Reviewer_qny2 · 2024-02-22
**This paper proposes a data augmentation technique to broaden the dataset scopes and  improve model performance without compromising patient confidentiality.**

**Rating:** 6
**Confidence:** 3

**Review:**

Advantages:
1. This paper proposes a novel method, which augments models using only the most predictive hypercube of synthetic data.
2. The motivation of the model is reasonable, and the application is important.

Disadvantages:
1. According to experimental results, the improvement from the model is relatively limited.
2. The author should introduce the concrete implementation process of the article in more detail.